# Self-supervised Graph Neural Networks via Low-Rank Decomposition

**Liang Yang** [1], **Runjie Shi** [1], **Qiuliang Zhang** [1], **Bingxin Niu** [1],
**Zhen Wang** [2], **Xiaochun Cao** [3], **Chuan Wang** [4*]
[1]School of Artificial Intelligence, Hebei University of Technology, Tianjin, China
[2]School of Artificial Intelligence, OPtics and ElectroNics (iOPEN),
School of Cybersecurity, Northwestern Polytechnical University, Xi'an, China
[3]School of Cyber Science and Technology,
Shenzhen Campus of Sun Yat-sen University, Shenzhen, China
[4]Institute of Information Engineering, CAS, Beijing, China
`yangliang@vip.qq.com, shirunjie2020@163.com, 3463194784@qq.com`
`niubingxin666@163.com, w-zhen@nwpu.edu.cn,`
`caoxiaochun@mail.sysu.edu.cn, wangchuan@iie.ac.cn`

## Abstract

Self-supervised learning is introduced to train graph neural networks (GNNs) by employing propagation-based GNNs designed for semi-supervised learning tasks. Unfortunately, this common choice tends to cause two serious issues. Firstly, global parameters cause the model lack the ability to capture the local property. Secondly, it is difficult to handle networks beyond homophily without label information. This paper tends to break through the common choice of employing propagation-based GNNs, which aggregate representations of nodes belonging to different classes and tend to lose discriminative information. If the propagation in each ego-network is just between the nodes from the same class, the obtained representation matrix should follow the low-rank characteristic. To meet this requirement, this paper proposes the Low-Rank Decomposition-based GNNs (LRD-GNN-Matrix) by employing Low-Rank Decomposition to the attribute matrix. Furthermore, to incorporate long-distance information, Low-Rank Tensor Decomposition-based GNN (LRD-GNN-Tensor) is proposed by constructing the node attribute tensor from selected similar ego-networks and performing Low-Rank Tensor Decomposition. The employed tensor nuclear norm facilitates the capture of the long-distance relationship between original and selected similar ego-networks. Extensive experiments demonstrate the superior performance and the robustness of LRD-GNNs.

## 1 Introduction

Graph Neural Networks (GNNs) have been successfully employed by many fields including computer vision (CV), natural language processing (NLP), information retrieval (IR), etc [1, 2, 3]. Vanilla GNNs, such as Graph Convolutional Network (GCN) [4] and Graph Attention Network (GAT) [5], are designed for semi-supervised node classification task, where part of nodes are labeled. To make models adaptive to networks beyond hompohily, flexible GNNs with more trainable parameters are proposed, such as JKNet [6], GCNII [7], H2GCN [8], FAGCN [9] and GPRGNN [10]. The role of label information is more critical for these flexible GNNs.

Inspired by the superior performance in CV and NLP, self-supervised learning, especially contrastive learning, is introduced to training GNNs. While a large number of methods are proposed, they can

---

[*]Corresponding author.

37th Conference on Neural Information Processing Systems (NeurIPS 2023).

be divided into two categories, i.e., contrastive and predictive models. The predictive models, such as Graph AutoEncoder (GAE) and its variational version (VAGE), MGAE [11], GALA [12] and GROVER, require data-label pairs, where the labels are self-generated from the data and significantly different for different tasks. Without the sensitive generated labels, contrastive models, which only require data-data pairs, attract more attention. The representative graph contrastive learning methods include DGI [13], GMI [14], MVGRL [15], GRACE [16], GCA [17] and BGRL [18].

Most existing graph self-supervised learning models employ vanilla GNNs designed for semi-supervised learning with global parameters as encoders. Especially, GCN with one or two layers, where only the feature mappings are trainable, is the most familiar choice. Unfortunately, this common choice tends to cause two serious issues. Firstly, global parameters cause the model lack the ability to capture the local property. For example, the mapping functions in GCN are shared by all nodes. Without the supervision from labels, the reliability of global parameters is weaker than that of the local ones. Secondly, it is difficult to handle networks beyond homophily. Most universal GNNs for networks beyond homophily introduce additional trainable parameters to distinguish neighbourhoods from different classes with the help of labels, such as GPRGNN and FAGNN. Without the label information, this process becomes complicated and fallible.

This paper tends to break through the common choice of employing propagation-based GNNs in self-supervised graph learning. Propagation-based GNNs seek node representation by averaging the representations of nodes, which may belong to different classes, from neighbourhoods, and thus tends to lose discriminative information. To overcome this issue, an intuitive and ideal way is to make the propagation in each ego-network just between the nodes from the same class. By this way, the obtained representation matrix for ego-network should follow the low-rank characteristic. To meet this requirement, this paper proposes the Low-Rank Decomposition-based GNNs (LRD-GNN) by employing Low-Rank Decomposition to the attribute matrix. By directly applying Low-Rank Matrix Decomposition to the attribute matrix of the ego-network, Low-Rank Matrix Decomposition-based GNN (LRD-GNN-Matrix) is given.

Furthermore, to incorporate long-distance information, Low-Rank Tensor Decomposition-based GNN (LRD-GNN-Tensor) is proposed by extending LRD-GNN-Matrix to tensor version. LRD-GNN-Tensor constructs the node attribute tensor by selecting similar ego-networks and splicing the attribute matrices of similar ego-networks into 3-way tensor. Motivated by the Tensor RPCA, tensor nuclear norm is defined as the average of the nuclear norm of all the frontal slices of discrete Fourier transformation of tensor along the 3-rd dimension. This tensor nuclear norm is equivalent to the matrix nuclear norm of the block circulant matrix of the tensor, which facilitates the capture of the long-distance relationship between original ego-network and selected similar ego-networks.

The main contributions of this paper are summarized as follows:

- We point out the necessity of designing specific encoder for self-supervised graph learning.
- We theoretically analyze the low-rank property of the representation matrix in ego-network.
- We proposed Low-Rank Decomposition-based GNNs (LRD-GNN) and its two instances, LRD-GNN-Matrix and LRD-GNN-Tensor.
- We experimentally evaluate the superior performance and the robustness of LRD-GNN.

## 2    Notations and Preliminaries

**Notations:** In this paper, boldface Euler script letters, *e.g.*, $\mathcal{A}$, are utilized to denote tensors. Matrices are denoted by boldface capital letters, *e.g.*, $\mathbf{A}$; vectors are denoted by boldface lowercase letters, *e.g.*, $\mathbf{a}$, and scalars are denoted by lowercase letters, *e.g.*, $a$. For a 3-way tensor $\mathcal{A}$, its $(i, j, k)$-th entry is represented as $\mathcal{A}_{ijk}$ or $a_{ijk}$ and use $\mathcal{A}(i, :, :)$, $\mathcal{A}(:, i, :)$ and $\mathcal{A}(:, :, i)$ to denote respectively the $i$-th horizontal, lateral and frontal slice. The frontal slice $\mathcal{A}(:, :, i)$ is denoted compactly as $\mathbf{A}^{(i)}$.

Let $\mathcal{G} = (\mathcal{V}, \mathcal{E})$ denote a graph with node set $\mathcal{V} = \{v_1, v_2, \cdots, v_N\}$ and edge set $\mathcal{E}$, where $N$ is the number of nodes. The topology of graph $\mathcal{G}$ can be represented by its adjacency matrix $\mathbf{A} = [a_{ij}] \in \{0, 1\}^{N \times N}$, where $a_{ij} = 1$ if and only if there exists an edge $e_{ij} = (v_i, v_j)$ between nodes $v_i$ and $v_j$. The degree matrix $\mathbf{D}$ is a diagonal matrix with diagonal element $d_i = \sum_{i=1}^{N} a_{ij}$ as the degree of node $v_i$. $\mathcal{N}(v_i) = \{v_j | (v_i, v_j) \in \mathcal{E}\}$ stands for the neighbourhoods of node $v_i$. Let $\mathcal{G}_i = (\mathcal{V}_i, \mathcal{E}_i)$ represents the ego-network around node $v_i$, where $\mathcal{V}_i = \mathcal{N}(v_i) \cup v_i$ and $\mathcal{E}_i$ denotes

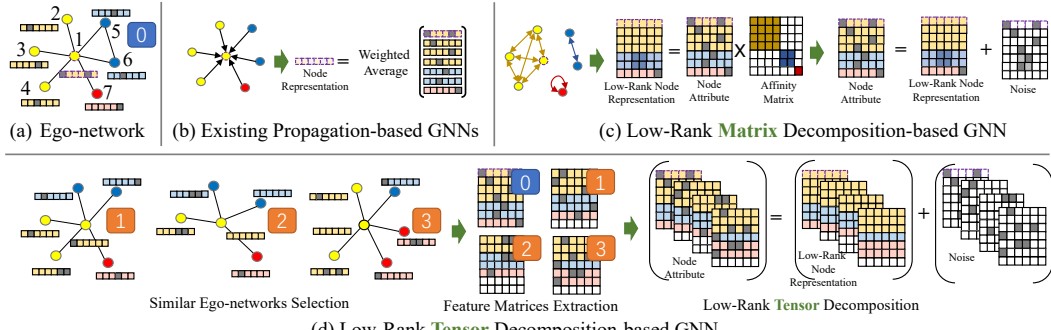

(a) Ego-network    (b) Existing Propagation-based GNNs          (c) Low-Rank **Matrix** Decomposition-based GNN

Similar Ego-networks Selection      Feature Matrices Extraction      Low-Rank **Tensor** Decomposition

(d) Low-Rank **Tensor** Decomposition-based GNN

Figure 1: Framework of the proposed Low-Rank Decomposition-based GNNs (LRD-GNNs) and comparison with existing GNNs. (a) One ego-network example. Colors of nodes stand for categories, while center node, i.e. node 1, is highlight with purple dashed border. (b) Existing propagation-based GNNs are equivalent to weighted averaging of neighbourhood nodes. (c) Low-Rank Matrix Decomposition-based GNN (LRD-GNN-Matrix). The propagations between nodes in the same class tend to make the embedding matrix of node is the ego-network be low-rank. To this end, low-rank matrix decomposition is applied to the feature matrix of nodes in the ego-network. (d) Low-Rank Tensor Decomposition-based GNN (LRD-GNN-Tensor). LRD-GNN-Tensor firstly selects ego-networks, which are similar to the ego-network example. Then, the low-rank tensor decomposition is applied to the tensor, which consists of the feature matrices from above selected ego-networks.

edges between nodes in $\mathcal{V}_i$. $\mathbf{X} \in \mathbb{R}^{N \times F}$ and $\mathbf{H} \in \mathbb{R}^{N \times F'}$ denote the collections of node attributes and representations with the $i^{th}$ rows, i.e., $\mathbf{x}_i \in \mathbb{R}^F$ and $\mathbf{h}_i \in \mathbb{R}^{F'}$, corresponding to node $v_i$, where $F$ and $F'$ stand for the dimensions of attribute and representation. For convenience, $\mathbf{X}_i \in \mathbb{R}^{(d_i+1) \times F}$ and $\mathbf{H}_i \in \mathbb{R}^{(d_i+1) \times F'}$ denote collections of attributes and representations of ego-network around $v_i$.

**Graph Neural Networks:** Most of the Graph Neural Networks (GNNs) follow an aggregation-combination strategy [3], where each node representation is iteratively updated by aggregating node representations in the local neighbourhoods and combining the aggregated representations with the node representation itself as

$$\bar{\mathbf{h}}_v^k = \text{AGGREGATE}^k \left( \{ \mathbf{h}_u^{k-1} | u \in \mathcal{N}(v) \} \right), \quad \mathbf{h}_v^k = \text{COMBINATE}^k \left( \mathbf{h}_v^{k-1}, \bar{\mathbf{h}}_v^k \right), \quad (1)$$

where $\bar{\mathbf{h}}_v^k$ stands for the aggregated representation from local neighbourhoods. Besides of the concatenation based implementation, such as GraphSAGE [19] and H2GCN [8], averaging (or summation) has been widely adopted to implement $\text{COMBINATE}^k(\cdot, \cdot)$, such as GCN [4], GAT [5], GIN [20], etc. Except for the MAX and LSTM implementations in GraphSAGE [19], most of the GNNs utilize averaging function to implement $\text{AGGREGATE}^k$. Therefore, they can be unified as

$$\mathbf{h}_v^k = \sigma \left( \left( c_{vv}^k \mathbf{h}_v^{k-1} + \sum_{u \in \mathcal{N}(v)} c_{uv}^k \mathbf{h}_u^{k-1} \right) \mathbf{W}^k \right), \quad (2)$$

where $\mathbf{W}^k$ represents the learnable parameters and $\sigma(\cdot)$ denotes the nonlinear mapping function.

## 3 Low-Rank Decomposition-based GNN

### 3.1 Motivations

Existing propagation-based GNNs tend to lose discriminative information. As shown in Figure 1(b) and Eq. (2), propagation-based GNNs seek node representation by averaging the representations of nodes from neighbourhoods. Unfortunately, neighbourhoods may belong to different classes from the center node. Thus, averaging representations of neighbourhood nodes may mix the representation of center node with representations of nodes from other classes. Especially, this issue is serious on networks with heterophily. The performance degradation of existing GNNs on networks with heterophily demonstrates the loss of discriminative information in propagation.

To alleviate this issue, an intuitive and ideal way is to make the propagation in each ego-network just between the nodes from the same class as shown in Figure 1(c). This can be formulated as

$$\mathbf{H}_i = \mathbf{X}_i \mathbf{P}_i, \quad (3)$$

where $\mathbf{X}_i \in \mathbb{R}^{(d_i+1)\times F}$ and $\mathbf{H}_i \in \mathbb{R}^{(d_i+1)\times F}$ denote the collections of node attributes and representations of ego-network around node $v_i$, i.e. $\mathcal{G}_i$. $\mathbf{P}_i \in \mathbb{R}^{(d_i+1)\times(d_i+1)}$ is the propagation matrix. If the propagation is only between the nodes from the same classes, $\mathbf{P}_i$ should be block diagonal matrix, whose rank is the same as the number of class in this ego-network. However, it is difficult to known the propagation matrix $\mathbf{P}_i$, since the classes of nodes are unknown in self-supervised learning task. Fortunately, it can be obtain that the rank of $\mathbf{H}_i$ is no more than the number of class in this ego-network, since the rank of $\mathbf{P}_i$ is the same as the number of class in this ego-network. Therefore, the matrix $\mathbf{H}_i$ should be low-rank.

## 3.2 Low-Rank Matrix Decomposition-based GNN

Based on the analysis in pervious subsection, this section proposes the Low-Rank Matrix Decomposition-based GNN (LRD-GNN-Matrix). Motivated by the Robust PCA (RPCA) [21], LRD-GNN-Matrix decomposes the attribute matrix $\mathbf{X}_i$ into low-rank $\mathbf{H}_i$ and sparse $\mathbf{S}_i$, i.e.,

$$\underset{\mathbf{H}_i, \mathbf{S}_i}{\operatorname{argmin}} rank(\mathbf{H}_i) + \lambda ||\mathbf{S}_i||_0 \quad s.t. \quad \mathbf{X}_i = \mathbf{H}_i + \mathbf{S}_i \tag{4}$$

where $rank(\mathbf{H})$ is the rank of the matrix $\mathbf{H}$, $||\mathbf{S}||_0$ is the number of non-zero elements in $\mathbf{S}$, i.e., $\ell_0$ norm, and $\lambda$ is a parameter for tradeoff between the two terms. Since the rank function and $\ell_0$ norm are nonconvex, we can alternatively minimize its convex surrogate as

$$\underset{\mathbf{H}_i, \mathbf{S}_i}{\operatorname{argmin}} ||\mathbf{H}_i||_* + \lambda ||\mathbf{S}_i||_1 \quad s.t. \quad \mathbf{X}_i = \mathbf{H}_i + \mathbf{S}_i \tag{5}$$

where $||\mathbf{H}||_*$ is the trace norm of $\mathbf{H}$, i.e., the sum of the singular values of $\mathbf{H}$. $||\mathbf{S}||_1$ is the $\ell_1$ norm, i.e. the sum of the absolute value of all elements in $\mathbf{S}$. By applying the Augmented Lagrangian Methods (ALM) [22], the constrained optimization problem in Eqs. (5) can be converted to

$$\mathcal{F}(\mathbf{H}_i, \mathbf{S}_i, \mathbf{Y}_i, \mu) = ||\mathbf{H}_i||_* + \lambda ||\mathbf{S}_i||_1 + <\mathbf{Y}_i, \mathbf{X}_i - \mathbf{H}_i - \mathbf{S}_i> + \frac{\mu}{2}||\mathbf{X}_i - \mathbf{H}_i - \mathbf{S}_i||_F^2, \tag{6}$$

where $\mathbf{Y}_i$ is the Lagrange multipliers for the constraint $\mathbf{X}_i = \mathbf{H}_i + \mathbf{S}_i$. $<\mathbf{Y}, \mathbf{X}> = tr(\mathbf{YX}')$ stands for the inner-product of matrix $\mathbf{Y}$ and $\mathbf{X}$. This objective function can be optimized via Alternating Direction Methods of Multipliers (ADMM) [23]. ADMM alternatively updates $\mathbf{H}_i$, $\mathbf{S}_i$ and $\mathbf{Y}_i$ by fixing others. The algorithm details are shown in Appendix. After the low-rank matrix $\mathbf{H}_i^*$ is obtained, the row in $\mathbf{H}_i^*$ corresponding to the node $v_i$ is extracted as $\mathbf{h}_i^*$ as the final representation of node $v_i$.

The proposed LRD-GNN-Matrix possesses two attractive characteristics: **(1) Parameter-free:** The above low-rank matrix decomposition is parameter-free. Thus, this procedure does not need the supervision label, and can be performed under self-supervised learning. **(2) Parallelization:** The proposed Low-rank GNNs only need the information of ego-network to obtain representation, and thus are easy to be parallelized. Thus, it is potential to be applied on large graphs.

## 3.3 Low-Rank Tensor Decomposition-based GNN

The proposed LRD-GNN-Matrix can partly alleviate the issue of discriminative information loss in propagation-based GNNs by low-rank representation pursuit. Unfortunately, it is also difficult to obtain robust and accurate node representations only using information in the ego-network, since the small number of nodes, which belong to the same class as center node, in the ego-network. Taking Figure 1(c) as an example, there are only 3 nodes in the ego-network, which belong to the same class as center node. This issue may be serious on networks with heterophily.

Many recent studies demonstrate that nodes with long distances from the center node can also provide additional information for center node representation. These distant nodes possess very different attributes from the center node, even they belong to the same class. Therefore, propagations with these distant nodes may benefit the robust representation of center node. To this end, this subsection enhances the LRD-GNN-Matrix by proposing Low-Rank Tensor Decomposition-based GNN (LRD-GNN-Tensor) as shown in 1(d). LRD-GNN-Tensor constructs the node attribute tensor by selecting similar ego-networks and splicing the attribute matrices of similar ego-networks into 3-way tensor. Analogous to the matrix case in Section 3.2, $\boldsymbol{\mathcal{X}}_i \in \mathbb{R}^{(d_i+1)\times F\times(M+1)}$ and $\boldsymbol{\mathcal{H}}_i \in \mathbb{R}^{(d_i+1)\times F\times(M+1)}$ denote the constructed node attribute tensor and representations tensor for ego-network around node

$v_i$, where $M$ is the number of selected similar ego-networks. To obtain robust node representation with the help of selected similar ego-networks, low-rank tensor decomposition is performed on $\boldsymbol{\mathcal{X}}_i$

$$\underset{\boldsymbol{\mathcal{H}}_i, \boldsymbol{\mathcal{S}}_i}{\operatorname{argmin}} ||\boldsymbol{\mathcal{H}}_i||_* + \lambda ||\boldsymbol{\mathcal{S}}_i||_1 \quad s.t. \quad \boldsymbol{\mathcal{X}}_i = \boldsymbol{\mathcal{H}}_i + \boldsymbol{\mathcal{S}}_i \tag{7}$$

The $\ell_1$ norm $|| \cdot ||_1$, which is the sum of the absolute value of all elements, can be directly extend from matrix to tensor.

Unfortunately, it is difficult to directly extend nuclear norm from matrix to tensor. Motivated by the Tensor RPCA (TRPCA) [24, 25], the tenor nuclear norm of a tensor $\boldsymbol{\mathcal{A}} \in \mathbb{R}^{n_1 \times n_2 \times n_3}$, denoted as $||\boldsymbol{\mathcal{A}}||_*$, is defined as the average of the nuclear norm of all the frontal slices of discrete Fourier transformation of $\boldsymbol{\mathcal{A}}$ along the 3-rd dimension $\bar{\boldsymbol{\mathcal{A}}} = \texttt{fft}(\boldsymbol{\mathcal{A}}, [], 3)$, i.e.$||\boldsymbol{\mathcal{A}}||_* = \frac{1}{n_3} \sum_{i=1}^{n_3} ||\bar{\mathbf{A}}^{(i)}||_*$, where $\bar{\mathbf{A}}^{(i)}$ denotes the frontal slice $\bar{\boldsymbol{\mathcal{A}}}(:,:,i)$. There exist an important property that the block circulant matrix can be mapped to a block diagonal matrix in the Fourier domain, i.e.,

$$(\mathbf{F}_{n_3} \otimes \mathbf{I}_{n_1}) \cdot \texttt{bcirc}(\boldsymbol{\mathcal{A}}) \cdot (\mathbf{F}_{n_3}^{-1} \otimes \mathbf{I}_{n_2}) = \bar{\mathbf{A}} \tag{8}$$

where $\mathbf{F}_{n_3}$ denotes discrete Fourier Transform matrix $\otimes$ stands for Kronecker product, and

$$\bar{\mathbf{A}} = \texttt{bdiag}(\bar{\boldsymbol{\mathcal{A}}}) = \begin{bmatrix} \bar{\boldsymbol{A}}^{(1)} & & & \\ & \bar{\boldsymbol{A}}^{(2)} & & \\ & & \ddots & \\ & & & \bar{\boldsymbol{A}}^{(n_3)} \end{bmatrix}, \quad \texttt{bcirc}(\boldsymbol{\mathcal{A}}) = \begin{bmatrix} \mathbf{A}^{(1)} & \mathbf{A}^{(n_3)} & \cdots & \mathbf{A}^{(2)} \\ \mathbf{A}^{(2)} & \mathbf{A}^{(1)} & \cdots & \mathbf{A}^{(3)} \\ \vdots & \vdots & \ddots & \vdots \\ \mathbf{A}^{(n_3)} & \mathbf{A}^{(n_3-1)} & \cdots & \mathbf{A}^{(1)} \end{bmatrix}.$$

According to this key property, the tensor nuclear norm is essentially equivalent to the matrix nuclear norm of the block circulant matrix, i.e.,

$$||\boldsymbol{\mathcal{A}}||_* = \frac{1}{n_3} \sum_{i=1}^{n_3} ||\bar{\mathbf{A}}^{(i)}||_* = \frac{1}{n_3} ||\bar{\mathbf{A}}||_* = \frac{1}{n_3} ||(\mathbf{F}_{n_3} \otimes \mathbf{I}_{n_1}) \cdot \texttt{bcirc}(\boldsymbol{\mathcal{A}}) \cdot (\mathbf{F}_{n_3}^{-1} \otimes \mathbf{I}_{n_2})||_* = \frac{1}{n_3} ||\texttt{bcirc}(\boldsymbol{\mathcal{A}})||_*.$$

If the selected $M$ similar ego-networks are the ego-networks around nodes $v_{i_1}, ..., v_{i_M}$, $\boldsymbol{\mathcal{X}}_i$ is constructed by splicing $\mathbf{X}_i$ and $\mathbf{X}_{i_1}, ..., \mathbf{X}_{i_M}$, and $\boldsymbol{\mathcal{H}}_i$ is constructed by splicing $\mathbf{H}_i$ and $\mathbf{H}_{i_1}, ..., \mathbf{H}_{i_M}$, then the tensor nuclear norm of $\boldsymbol{\mathcal{H}}_i$ is

$$||\boldsymbol{\mathcal{H}}_i||_* = \frac{1}{M+1} ||\texttt{bcirc}(\boldsymbol{\mathcal{H}}_i)||_* = \frac{1}{M+1} \left\| \begin{bmatrix} \mathbf{H}_i & \mathbf{H}_{i_M} & \cdots & \mathbf{H}_{i_1} \\ \mathbf{H}_{i_1} & \mathbf{H}_i & \cdots & \mathbf{H}_{i_2} \\ \vdots & \vdots & \ddots & \vdots \\ \mathbf{H}_{i_M} & \mathbf{H}_{i_{M-1}} & \cdots & \mathbf{H}_{i,} \end{bmatrix} \right\|_* \tag{9}$$

which can capture the long-distance relationship between original ego-networ $\mathcal{G}_i$ and selected similar ego-networks $\mathcal{G}_{i_1}, ..., \mathcal{G}_{i_M}$. Therefore, compared to LRD-GNN-Matrix, LRD-GNN-Tensor can exploit long-distance information, and thus obtain robust and accurate node representation. The detailed algorithm to optimize Eq. (7) is given in Appendix.

## 4 Experiments

**Datasets.** Our experiments are conducted on 12 commonly used benchmark datasets, including 6 homophilic graph datasets (i.e., Cora, CiteSeer, PubMed, Wiki-CS, Amazon Computers and Amazon Photo [26, 27, 28]) and 6 heterophilic graph datasets (i.e., Chameleon, Squirrel, Actor, Cornell, Texas, and Wisconsin [29]). The statistics of datasets are summarized in Table 1.

**Cora, CiteSeer and PubMed** [26] are three citation network datasets, where nodes indicate a paper and each edge indicates a citation relationship between two papers. The labels are the research topic of papers. **Wiki-CS** [27] is a reference network constructed based on Wikipedia. The nodes correspond to articles about computer science and edges are hyperlinks between the articles. Nodes are labeled with ten classes each representing a branch of the field. **Amazon Computers and Amazon Photo** [28] are two co-purchase networks from Amazon. In these networks, each node indicates a good, and each edge indicates that two goods are frequently bought together. The labels are the category of goods. **Cornell, Texas and Wisconsin** [29] are three web page networks from computer science departments of diverse universities, where nodes are web pages and edges are hyperlinks

Table 1: Statistics of datasets

| Dataset | Cora | CiteSeer | PubMed | Wiki-CS | Computers | Photo | Chameleon | Squirrel | Actor | Cornell | Texas | Wisconsin |
|---|---|---|---|---|---|---|---|---|---|---|---|---|
| # Nodes | 2,708 | 3,327 | 19,717 | 11,701 | 13,752 | 7,650 | 2,277 | 5,201 | 7,600 | 183 | 183 | 251 |
| # Edges | 5,429 | 4,732 | 44,338 | 216,123 | 245,861 | 119,081 | 36,101 | 217,073 | 33,544 | 295 | 309 | 499 |
| # Features | 1,433 | 3,703 | 500 | 300 | 767 | 745 | 2,325 | 2,089 | 932 | 1,703 | 1,703 | 1,703 |
| # Classes | 7 | 6 | 3 | 10 | 10 | 8 | 5 | 5 | 5 | 5 | 5 | 5 |

Table 2: Results in terms of classification accuracies (in percent ± standard deviation) on homophilic benchmarks. The best and runner-up results are highlighted with **bold** and underline, respectively.

| Methods | Cora | CiteSeer | PubMed | Wiki-CS | Computers | Photo |
|---|---|---|---|---|---|---|
| GCN | 81.50±1.30 | 70.30±0.28 | 78.80±2.90 | 76.89±0.37 | 86.34±0.48 | 92.35±0.25 |
| GAT | 82.80±1.30 | 71.50±0.49 | 78.50±0.27 | 77.42±0.19 | 87.06±0.35 | 92.64±0.42 |
| MLP | 56.11±0.34 | 56.91±0.42 | 71.35±0.05 | 72.02±0.21 | 73.88±0.10 | 78.54±0.05 |
| JKNet | 81.10±0.00 | 69.80±0.36 | 78.10±0.24 | 79.52±0.21 | 85.28±0.72 | 92.68±0.13 |
| H2GCN | 80.23±0.20 | 69.97±0.66 | 78.79±0.30 | 79.73±0.13 | 84.32±0.52 | 91.86±0.27 |
| FAGCN | 77.80±0.66 | 69.81±0.80 | 76.74±0.66 | 74.34±0.53 | 83.51±1.04 | 92.72±0.22 |
| GPR-GNN | 80.55±1.05 | 68.57±1.22 | 77.02±2.59 | 79.82±0.35 | 86.71±1.82 | 92.93±0.26 |
| DeepWalk | 69.47±0.55 | 58.82±0.61 | 69.87±1.25 | 74.35±0.06 | 85.68±0.06 | 89.44±0.11 |
| node2vec | 71.24±0.89 | 47.64±0.77 | 66.47±1.00 | 71.79±0.05 | 84.39±0.08 | 89.67±0.12 |
| GAE | 71.07±0.39 | 65.22±0.43 | 71.7310.92 | 70.15±0.01 | 85.27±0.19 | 91.62±0.13 |
| VGAE | 79.81±0.87 | 66.75±0.37 | 77.16±0.31 | 76.63±0.19 | 86.37±0.21 | 92.20±0.11 |
| DGI | 82.29±0.56 | 71.49±0.14 | 77.43±0.84 | 75.73±0.13 | 84.09±0.39 | 91.49±0.25 |
| GMI | 82.51±1.47 | 71.56±0.56 | 79.83±0.90 | 75.06±0.13 | 81.76±0.52 | 90.72±0.33 |
| MVGRL | 83.03±0.27 | **72.75±0.46** | 79.63±0.38 | 77.97±0.18 | 87.09±0.27 | 92.01±0.13 |
| GRACE | 80.08±0.53 | 71.41±0.38 | 80.15±0.34 | 79.16±0.36 | 87.21±0.44 | 92.65±0.32 |
| GCA | 80.39±0.42 | 71.21±0.24 | **80.37±0.75** | 79.35±0.12 | 87.84±0.27 | 92.78±0.17 |
| BGRL | 81.08±0.17 | 71.59±0.42 | 79.97±0.36 | 78.74±0.22 | 88.92±0.33 | 93.24±0.29 |
| LRD-GNN-Matrix | 82.10±0.24 | 71.91±0.56 | 78.50±1.20 | 80.19±0.18 | 87.15±0.19 | 92.31±0.17 |
| LRD-GNN-Tensor | **83.74±0.61** | 72.05±0.69 | 79.71±0.75 | **81.43±0.13** | **89.60±0.18** | **93.26±0.15** |

between two web pages. The labels are types of web pages. **Chameleon and Squirrel** [29] are two Wikipedia networks where nodes denote web pages in Wikipedia and edges denote links between two pages. The labels stand for the average traffic of the web page.

For Cora, CiteSeer, and PubMed datasets, we adopt the public splits with 20 labeled nodes per class for training, 500 nodes for validation and 1000 nodes for testing. For Wiki-CS, Computers and Photo datasets, we randomly split all nodes into three parts, i.e., 10% nodes for training, 10% nodes for validation and the remaining 80% nodes for testing. The performance on heterophilic datasets is evaluated on the commonly used 48%/32%/20% training/validation/testing.

**Baselines.** To verify the superiority of the proposed LRD-GNN , We compare it with four groups of baseline methods: (1) The multiple layer perception (MLP) and classic GNN models for node classification task including vanilla GCN [4] and GAT [5]; (2) GNN models designed for alleviating over-smoothing issue or networks with heterophily including JKNet [6], GPR-GNN [10], FAGCN [9] and H2GCN [8]; (3) Conventional self-supervised graph representation learning methods including DeepWalk [30], node2vec [31], GAE and VGAE [11]; (4) Contrastive self-supervised baselines including DGI [13], GMI [14], MVGRL [15], GRACE [16], GCA [17] and BGRL [18].

**Experimental details.** All methods were implemented in Pytorch with Adam Optimizer. We run 10 times of experiments and report the averaged test accuracy with standard deviation. All the parameters of baselines are tuned to get preferable performance in most situations or the same as authors' original implementations. The hyper-parameter search space is: learning rate in {0.1, 0.05, 0.01}, dropout in {0.2, 0.3, 0.4}. Besides, early stopping with a patience of 200 epochs and L2 regularization with coefficient in {1E-2, 5E-3, 1E-3} are employed to prevent overfitting.

## 4.1 Experimental Results

### 4.1.1 Evaluation on node classification task

**Performance Comparison.** The mean classification accuracy with the standard deviation on 6 homophilic datasets and 6 heterophilic datasets are presented in Table 2 and Table 3, respectively. We

Table 3: Results in terms of classification accuracies (in percent ± standard deviation) on heterophilic benchmarks. The best and runner-up results are highlighted with **bold** and underline, respectively.

| Methods | Chameleon | Squirrel | Actor | Cornell | Texas | Wisconsin |
|---|---|---|---|---|---|---|
| GCN | 59.63±2.32 | 36.28±1.52 | 30.83±0.77 | 57.03±3.30 | 60.00±4.80 | 56.47±6.55 |
| GAT | 56.38±2.19 | 32.09±3.27 | 28.06±1.48 | 59.46±3.63 | 61.62±3.78 | 54.71±6.87 |
| MLP | 46.91±2.15 | 29.28±1.33 | 35.66±0.94 | 81.08±7.93 | 81.62±5.51 | 84.31±3.40 |
| JKNet | 58.31±2.76 | 42.24±2.11 | 36.47±0.51 | 56.49±3.22 | 65.35±4.68 | 51.37±3.21 |
| H2GCN | 59.39±1.98 | 37.90±2.02 | 35.86±1.03 | 82.16±4.80 | 84.86±6.77 | 86.67±4.69 |
| FAGCN | 63.44±2.05 | 41.17±1.94 | 36.81±0.26 | 81.35±5.05 | 84.32±6.02 | 83.33±2.01 |
| GPR-GNN | 61.58±2.24 | 46.65±1.81 | 35.27±1.04 | 81.89±5.93 | 83.24±4.95 | 84.12±3.45 |
| DeepWalk | 47.74±2.05 | 32.93±1.58 | 22.78±0.64 | 39.18±5.57 | 46.49±6.49 | 33.53±4.92 |
| node2vec | 41.93±3.29 | 22.84±0.72 | 28.28±1.27 | 42.94±7.46 | 41.92±7.76 | 37.45±7.09 |
| GAE | 33.84±2.77 | 28.03±1.61 | 28.03±1.18 | 58.85±3.21 | 58.64±4.53 | 52.55±3.80 |
| VGAE | 35.22±2.71 | 29.48±1.48 | 26.99±1.56 | 59.19±4.09 | 59.20±4.26 | 56.67±5.51 |
| DGI | 39.95±1.75 | 31.80±0.77 | 29.82±0.69 | 63.35±4.61 | 60.59±7.56 | 55.41±5.96 |
| GMI | 46.97±3.43 | 30.11±1.92 | 27.82±0.90 | 54.76±5.06 | 50.49±2.21 | 45.98±2.76 |
| MVGRL | 51.07±2.68 | 35.47±1.29 | 30.02±0.70 | 64.30±5.43 | 62.38±5.61 | 62.37±4.32 |
| GRACE | 48.05±1.81 | 31.33±1.22 | 29.01±0.78 | 54.86±6.95 | 57.57±5.68 | 50.00±5.83 |
| GRACE-FA | 52.68±2.14 | 35.97±1.20 | 32.55±1.28 | 67.57±4.98 | 64.05±7.46 | 63.73±6.81 |
| GCA | 49.80±1.81 | 35.50±0.91 | 29.65±1.47 | 55.41±4.56 | 59.46±6.16 | 50.78±4.06 |
| BGRL | 47.46±2.74 | 32.64±0.78 | 29.86±0.75 | 57.30±5.51 | 59.19±5.85 | 52.35±4.12 |
| LRD-GNN-Matrix | 60.71±2.21 | 47.64±1.21 | **37.22±0.83** | 78.48±0.36 | 86.12±2.86 | 83.13±1.90 |
| LRD-GNN-Tensor | **66.27±1.27** | **55.91±0.91** | 37.07±0.83 | **84.73±2.83** | **87.03±3.97** | **86.86±3.04** |

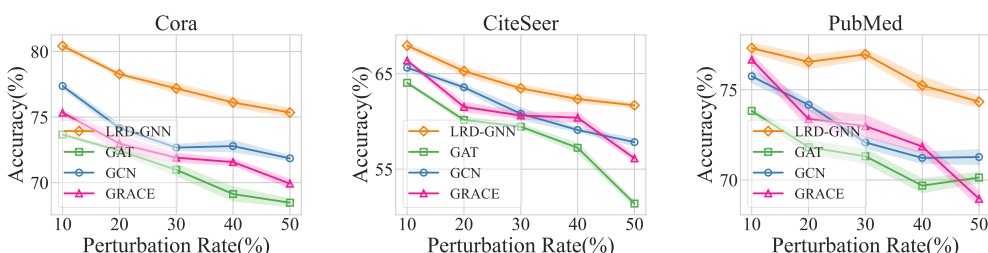

Figure 2: Node classification performance on graphs with randomly adding noisy edges.

compare the proposed LRD-GNN-Matrix and its tensor version LRD-GNN-Tensor with baselines. First of all, we observe that LRD-GNN-Tensor outperforms all baseline methods in 10 out of 12 benchmarks and achieves competitive results on the rest 2 benchmarks.

The superior performance indicates the tensor low-rank decomposition can universally benefit representation learning, since low-rank structure widely exists in diverse real-world graph data. And the construction of tensors effectively captures the connections among similar nodes. Note that LRD-GNN outperforms state-of-the-art deep model JKNet on both homophilic and heterophilic datasets. This demonstrates that shallow-layer information is actually quite abundant for extracting node representation as for LRD-GNN . Additionally, compared with GPR-GNN, FAGNN and H2GCN which are all the GNNs designed for processing datasets with heterophily, we observe that LRD-GNN achieves new state-of-the-art results on all heterophilic datasets. This verifies that the low-rank structure remains the heterophilic information from the ego-networks instead of the impairing heterophilic information as processing by averaging operation.

In Table 3, we find that LRD-GNN significantly outperforms conventional and contrastive self-supervised methods. The main reason is that these self-supervised methods constantly smooth the representations along heterophilic edges, which destroys the low-rank structure and makes the representations indistinguishable. In contrast, our method overcomes these issues via low-rank tensor decomposition, and learns more expressive representations. Besides, it can be observed that LRD-GNN-Tensor outperforms LRD-GNN-Matrix in most cases, which indicates that combining global information from distant nodes with the local is necessary. These results suggest that by adopting the low-rank decomposition in GNNs, our proposed LRD-GNN is more effective and universal than the previous models on processing datasets with both homophily and heterophily for node classification.

The following experiments are all based on LRD-GNN-Tensor, which is abbreviated as LRD-GNN.

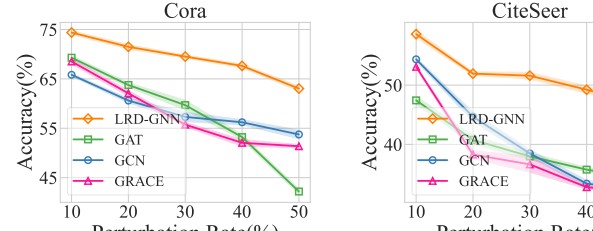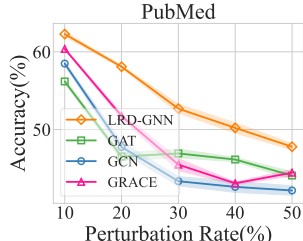

Figure 3: Node classification performance on graphs with randomly adding noisy attributes.

Table 4: Performance of LRD-GNN under different tensor construction methods

| Similar Nodes | Data Variants | Cora | CiteSeer | Chameleon | Cornell | Texas |
|---|---|---|---|---|---|---|
| Attribute | $X, AX, A^2X$ | 83.74±0.61 | 72.05±0.69 | 65.92±1.20 | 88.95±3.11 | 88.42±1.29 |
| | $X$ | 83.55±0.27 | 71.59±0.91 | 56.29±0.67 | 88.15±2.94 | 87.82±1.28 |
| | $X, AX$ | 83.62±0.54 | 71.67±0.60 | 53.46±1.68 | 87.89±2.11 | 87.36±3.68 |
| | $AX$ | 82.66±0.71 | 71.30±0.74 | 47.32±2.66 | 71.31±1.85 | 72.78±2.37 |
| | $AX, A^2X$ | 80.32±0.49 | 70.06±0.65 | 63.07±1.09 | 73.05±2.36 | 73.68±1.66 |
| Structure | $X$ | 81.25±0.42 | 71.65±0.68 | 55.26±1.29 | 84.73±1.97 | 85.78±1.29 |
| | $X, AX$ | 80.73±0.41 | 71.03±0.75 | 45.68±1.35 | 84.21±1.66 | 83.68±1.05 |
| | $X, AX, A^2X$ | 78.20±0.65 | 69.21±0.64 | 64.27±1.39 | 83.15±2.68 | 84.73±3.06 |

**Visualization.** To provide an intuitive interpretation, we apply t-SNE to visualize the node embeddings obtained by GCN, GAT and LRD-GNN on four datasets. As shown in Figure 4, the clusters of embeddings of nodes from different classes are marked with various colors. By visually analyzing the clusters and distribution of the node classification results, we can find the characteristics of the corresponding models. The clusters of embeddings of different classes processed by GCN are overlapped, which demonstrates that GCN tends to be under-fitting. The clusters of embeddings obtained by GAT are also irregular and sharp, especially the poor performance on Chameleon which indicates that simply aggregating information does not work on homophilic graphs. While the clusters of embedding obtained by LRD-GNN are more regular and the nodes with the same label exhibit spatial clustering, which shows the power of LRD-GNN.

### 4.1.2 Robustness Analysis

In this experiment, we investigate the robustness of LRD-GNN on graph data. We perturb graph structure and input node attributes by randomly adding noisy edges and attributes respectively, and test the node classification accuracy on the representations learned from perturbed graphs. We compare the performance of LRD-GNN with GCN, GAT and GRACE on Cora, CiteSeer and Pubmed datasets. The classification results under different perturbation rates are shown in Figure 2 and Figure 3. From Figure 2, we can see that LRD-GNN consistently outperforms the other three methods under different perturbation rates. Specifically, the proposed LRD-GNN obtains node representations without propagation and the graph topology is only used to form the ego-network. However, other methods need to propagate the information based on topology, which indicates that they are more likely to be affected by corrupted topology information. Therefore, LRD-GNN tends to be more robust to topology noises than the existing GNNs. Figure 3 reports that LRD-GNN is also superior to other baselines under attributes perturbation, which can be attributed to the denoising ability of low-rank decomposition and recovery in LRD-GNN . These experimental results demonstrate the strong robustness of LRD-GNN against random attacks on graph topology and node attributes.

### 4.1.3 Ablation Study

To analyze the effectiveness of different tensor construction methods in LRD-GNN , we conduct experiments on several variants of tensor. We compare the performance of LRD-GNN under different tensor construction methods, and node classification results with several variants are shown in Table 4. We mainly investigate methods of constructing tensors from topology and attribute two perspectives: (1) select nodes with similar attributes; (2) select nodes with similar local structures, and then use

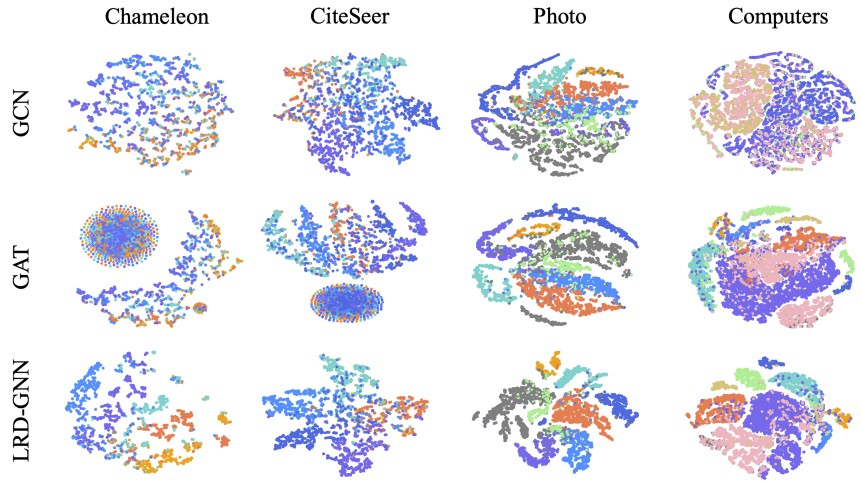

Figure 4: The visualization for node representations obtained by GCN, GAT and LRD-GNN .

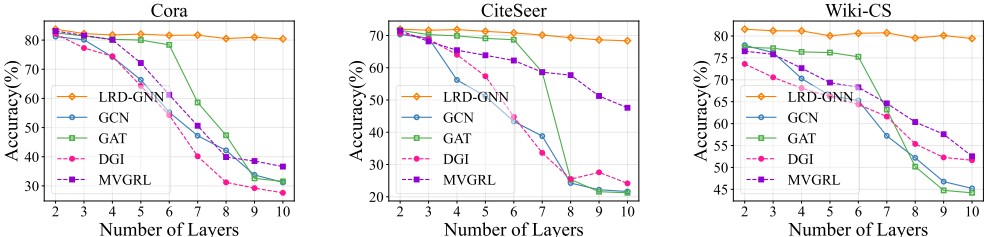

Figure 5: Node classification results with various model depths on Cora, Citeseer and Wiki-CS.

these nodes' ego-network matrix in different orders (i.e., $X$, $AX$, $A^2X$) to construct tensors, where each ego-network matrix forms a slice of the target tensor.

From Table 4, we can find that under the same configuration, node embeddings obtained from tensors constructed by nodes with similar attributes are more expressive than tensors constructed by nodes with similar local structures, both on the homophilic graphs (Cora and CiteSeer) and heterophilic graphs (Chameleon, Cornell and Texas). When higher-order information (i.e., $A^2X$) is utilized, the performance on heterophilic datasets is significantly improved. However, the variant of tensor which contains $AX$ is not competitive enough compared to other variants on heterophilic graphs, which indicates that higher-order information is important for representation learning on heterophilic graphs. In addition, we can observe that tensors that preserve the original node attributes outperform tensors formed only by node representations obtained by propagation, which indicates that original node attributes play a prominent role.

### 4.1.4 Over-smoothing Problem Analysis

To validate whether LRD-GNN can alleviate the over-smoothing problem, we compare the performance of LRD-GNN with several representative self-supervised and semi-supervised GNNs under different model depth. We regard the process of obtaining node representations via TensorRPCA as a layer of LRD-GNN framework. The node classification results on Cora, CiteSeer and Wiki-CS are shown in Figure 5. It can be seen that GCN achieves competitive performance at two layers. As the number of layers increases, the performance of GCN drops rapidly, which indicates that GCN suffers from over-smoothing seriously. GAT alleviates the over-smoothing issue by introducing irrelevant multi-channel propagations. Unfortunately, it also becomes over-smoothing after few layers. DGI and MVGRL, which adopt GCN layer as encoder, also tend to be over-smoothing as depth increases. Instead, the results of LRD-GNN are stable and higher than other methods on different types of networks. The reasons are two-folds: one is that the proposed LRD-GNN obtains node representations without propagation and the graph topology is only used to form the ego-network, which can prevent node representations from being too similar. Another is that the low-rank decom-

position in LRD-GNN reduces the impact of noise, which further keeps node representations from becoming indistinguishable. Through these two designs, when the model goes deep, the performance of LRD-GNN is significantly better than the above self-supervised and semi-supervised methods, which indicates that LRD-GNN has a good capability to alleviate over-smoothing.

## 5 Conclusions

This paper breaks through the common choice of employing propagation-based GNNs, which aggregate representations of nodes belonging to different classes and tend to lose discriminative information. Firstly, this paper theoretically shows that the obtained representation matrix should follow the low-rank characteristic. Then, to meet this requirement, this paper proposes the Low-Rank Decomposition-based GNNs (LRD-GNN) by employing Low-Rank Decomposition to the attribute matrix. Furthermore, to incorporate long-distance information, Low-Rank Tensor Decomposition-based GNN (LRD-GNN-Tensor) is proposed by constructing the node attribute tensor from selected similar ego-networks and performing Low-Rank Tensor Decomposition. Extensive experiments demonstrate the superior performance and the robustness to noises of the proposed LRD-GNNs.

## Acknowledgment

This work was supported in part by the National Natural Science Foundation of China (No. U22B2036, 62376088, 61972442, 62102413, U1936210, U1936208, 11931915), in part by the National Science Fund for Distinguished Young Scholarship of China (No. 62025602), in part by the Natural Science Foundation of Hebei Province of China under Grant F2020202040, in part by the Fok Ying-Tong Education Foundation China (No. 171105), and in part by the Tencent Foundation and XPLORER PRIZE.

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

# A  Algorithm Description of ADMM

We use Alternating Direction Methods of Multipliers (ADMM) to solve convex optimization problem in RPCA and TensorRPCA , and the details are shown in Algorithm1 and Algorithm2 which are named Matrix ADMM and Tensor ADMM, respectively.

---

**Algorithm 1:** Matrix ADMM

**Input:** matrix data $X$, parameter $\lambda$.
**Initialization:** $H_0 = S_0 = Y_0 = 0, \rho = 1.1, \mu_0 = 1e - 3, \epsilon = 1e - 8$.
**while** not converged **do**

> % Singular Value Thresholding %;
> 1. Update $H_{k+1}$ by $H_{k+1} = \arg\min_{H} \|H\|_* + \frac{\mu_k}{2}\|H + S_k - X + \frac{Y_k}{\mu_k}\|_F^2$ ;
>
> % Soft-Thresholding %;
> 2. Update $S_{k+1}$ by $S_{k+1} = \arg\min_{S} \lambda\|S\|_1 + \frac{\mu_k}{2}\|H_{k+1} + S - X + \frac{Y_k}{\mu_k}\|_F^2$;
>
> 3. $Y_{k+1} = H_k + \mu_k(H_{k+1} - S_{k+1} - X)$ ;
>
> 4. Update $\mu_{k+1}$ by $\mu_{k+1} = \rho\mu_k$ ;
>
> 5. Check the convergence conditions
>
> $\|H_{k+1} - H_k\|_\infty \le \epsilon, \|S_{k+1} - S_k\|_\infty \le \epsilon, \|H_{k+1} - S_{k+1} - X\|_\infty \le \epsilon$;

**end**

---

**Algorithm 2:** Tensor ADMM

**Input:** tensor data $\mathcal{X}$, parameter $\lambda$.
**Initialization:** $\mathcal{H}_0 = \mathcal{S}_0 = \mathcal{Y}_0 = 0, \rho = 1.1, \mu_0 = 1e - 2, \epsilon = 1e - 5$.
**while** not converged **do**

> % Singular Value Thresholding %;
> 1. Update $\mathcal{H}_{k+1}$ by $\mathcal{H}_{k+1} = \arg\min_{\mathcal{H}} \|\mathcal{H}\|_* + \frac{\mu_k}{2}\|\mathcal{H} + \mathcal{S}_k - \mathcal{X} + \frac{\mathcal{Y}_k}{\mu_k}\|_F^2$ ;
>
> % Soft-Thresholding %;
> 2. Update $\mathcal{S}_{k+1}$ by $\mathcal{S}_{k+1} = \arg\min_{\mathcal{S}} \lambda\|\mathcal{S}\|_1 + \frac{\mu_k}{2}\|\mathcal{H}_{k+1} + \mathcal{S} - \mathcal{X} + \frac{\mathcal{Y}_k}{\mu_k}\|_F^2$;
>
> 3. $\mathcal{Y}_{k+1} = \mathcal{H}_k + \mu_k(\mathcal{H}_{k+1} - \mathcal{S}_{k+1} - \mathcal{X})$ ;
>
> 4. Update $\mu_{k+1}$ by $\mu_{k+1} = \min(\rho\mu_k, \mu_{\max})$ ;
>
> 5. Check the convergence conditions
>
> $\|\mathcal{H}_{k+1} - \mathcal{H}_k\|_\infty \le \epsilon, \|\mathcal{S}_{k+1} - \mathcal{S}_k\|_\infty \le \epsilon, \|\mathcal{H}_{k+1} - \mathcal{S}_{k+1} - \mathcal{X}\|_\infty \le \epsilon$;

**end**

---

# B  Select Similar Ego-networks

In section 3.3, we enhance LRD-GNN-Matrix to tensor version LRD-GNN-Tensor. The ego-networks around nodes which are similar with target node are selected to construct tensor. We evaluate similarity from two perspectives. (1) We select nodes which have similar attributes with target node. For instance, the Cosine Similarity is used to measure the similarity of attributes between nodes. (2) We select nodes which have similar local structures with target node. We use the Shannon entropy value of ego-network to measure local structure similarity. For the ego-network $\mathcal{G}_i = (\mathcal{V}_i, \mathcal{E}_i)$ around $v_i$, the Shannon entropy value $H(\mathcal{G}_i)$ is defined as

$$H(\mathcal{G}_i) = -\sum_{v \in \mathcal{V}_i} P(v) \log P(v) \tag{10}$$

where $P(v)$ is the probability of the random walking visiting $v$ in ego-network. Then, we choose several nodes' ego-networks that are close to the Shannon entropy value of the central node to construct tensor.

Besides, to combine information from different orders, we also utilize the selected nodes' ego-networks after propagation(i.e., $AX, A^2X$) as part of the tensor. The specific methods for constructing tenosr and the performance have been presented in section 4.1.3 Ablation Study.

