# OpenReview forum: "Self-supervised Graph Neural Networks via Low-Rank Decomposition"
_NeurIPS.cc/2023/Conference — NeurIPS 2023 poster_

### Official Review · Reviewer_KVoZ · 2023-07-03

**Soundness:** 3 good
**Presentation:** 3 good
**Contribution:** 3 good
**Rating:** 7
**Confidence:** 4

**Summary:**

The paper argues that when dealing with self-supervised learning tasks, utilizing propagation-based GNNs as encoder inevitably encounters two serious issues, i.e., failing to capture local property due to global parameters and lacking the ability on handling homogenous networks without label information. To this end, the authors introduce a different perspective by replacing the propagation-based GNNs with low-rank decomposition-based GNNs. The authors formulate the low-rank decomposition-based GNNs by employing low-rank decomposition to the attribute matrix. Besides, considering that networks require long-distance information for representation, the authors introduce the tensor-based formulation and construct the node attribute tensor from selected similar ego-networks.

**Strengths:**

- The paper provides a novel perspective to tackle GNNs encoder for self-supervised learning.
- The proposed low-rank decomposition-based GNNs effectively solve the serious issues introduced by employing propagation-based GNNs.
- The consideration of incorporating long-distance information successfully captures the long-distance relationships between original and selected similar ego-networks.
- The experiments show the superiority and generalization of the proposed low-rank tensor decomposition-based GNNs. The ablation study and experimental analysis illustrate the success in a complete view.


**Weaknesses:**

- The reason why the low-rank decomposition-based GNNs preserve local information is not presented clearly.
- Can the authors clarify the differences between networks beyond homophily and network with heterophily?
- How can the proposed model be enhanced in semi-supervised tasks by using the node labels.


**Questions:**

see Weaknesses.

---

> ### Author Rebuttal · Authors · 2023-08-09
>
> **Q1. The reason why the low-rank decomposition-based GNNs preserve local information is not presented clearly.**
>
> R1. The characteristic of local information preservation is from both the matrix/tensor construction and low-rank decomposition. Firstly, both the matrix and the slice of the tensor, which will be decomposed, are constructed from the attributes of nodes in the local ego network. This leads the obtained representation to learn from local information. Secondly, the low-rank property is the requirement of the local propagation between nodes with the same labels. Thus, the low-rank decomposition, which seeks low-rank representation, essentially performs local propagation. Therefore, the low-rank decomposition-based GNNs possess the potential of preserving local information.
>
> ---
>
> **Q2. Can the authors clarify the differences between networks beyond homophily and network with heterophily?**
>
> R2. The homophily rate is a measurement to describe the proportion of linked nodes belonging to the same classes. Vanilla GNNs perform well on networks with high homophily rates, i.e. homophilic networks. The network with heterophily are the networks with low homophily rates, while the networks beyond homophily are the networks, whose homophily rates are not high. Therefore, networks with heterophily belong to networks beyond homophily.
>
> ---
>
> **Q3. How can the proposed model be enhanced in semi-supervised tasks by using the node labels?**
>
> R3. The proposed model can be enhanced by constructing the matrix and tensor with the help of node labels. The current self-supervised model constructs the matrix and tensor, which will be decomposed, using the ego-network and similar ego-networks without any supervision. When the node labels are available, the matrix and tensor constructions can be improved according to the predicted node labels. Specifically, the labels of nodes, which are unlabelled, can be predicted using the existing GNNs. Then, the matrix can be constructed using the nodes in the ego-network, which share the same predicted labels as center nodes. And the tensor can be constructed using the ego-networks, whose center nodes’ labels are the same. By employing the predicted node label, the nodes in the constructed matrix and tensor may belong to the same classes with higher probability, thus their ranks tend to be much lower. This facilitates the following decomposition. Therefore, the model can be enhanced with the help of node labels.

---

### Official Review · Reviewer_EJvt · 2023-07-06

**Soundness:** 3 good
**Presentation:** 3 good
**Contribution:** 4 excellent
**Rating:** 7
**Confidence:** 4

**Summary:**

The encoder for self-supervised graph neural networks is investigated.  The authors identify the weaknesses of capturing local property and handling heterophilic networks in existing propagation-based encoder. They observe that the obtained node representations possess low-rank characteristics and tend to meet it using the low-rank matrix factorization. Besides, to extend to incorporate global information, attribute tensor is constructed and low-rank tensor factorization is applied. Experiments shows its effectiveness and robustness to noises, especially on networks with heterophily.

**Strengths:**

•	The investigated encoder issues are significant to self-supervised graph neural network. Especially, the global parameter and the lack of supervision information are critical for encoder choice.

•	The employed low-rank decomposition algorithms are novel to GNNs. The decomposition to local attribute matrix facilitates the local structure capture and the requirements to label.

•	The performance enhancement on heterophilic networks is remarkable. Experiments on robustness and preventing oversmoothing are convincing.

•	The presentation is clear. The figure of the proposed framework is elaborate to show the process. The figure of visualization demonstrates the discriminability of node representations.


**Weaknesses:**

•	The relationship between the proposed method and existing self-supervised GNN is not clear. Existing self-supervised learning relays on encoder for supervised learning and an objective function. However, the proposed method does not need objective function. Therefore, it is important to find their connections.

•	It is difficult for readers, who are not familiar with low-rank, to get the key points in motivations and details of the proposed methods. It is better to review the concepts of low-rank and self-supervised in preliminaries section or appendix.

•	There are some typos. For example, the is a nuclear norm missing in the formula between lines 162 and 163.

•	The performance on preventing over-smoothing issue is compared against semi-supervised GNNs, i.e., GCN and GAT. The self-supervised GNNs should be compared with.

•	The proposed LRD-GNN is a non-parameter model, which may benefit the task of self-supervised learning. However, how can it be improved if the labels are available.


**Questions:**

Refer to Weaknesses.

**Limitations:**

None.

---

> ### Author Rebuttal · Authors · 2023-08-09
>
> ***Q1.  The relationship between the proposed method and existing self-supervised GNN is not clear. Existing self-supervised learning relays on encoder for supervised learning and an objective function. However, the proposed method does not need objective function. Therefore, it is important to find their connections.***
>
> R1. Thanks for your advice. Unfortunately, it is difficult to point out their detailed connections, since they are very different from design principal. Existing self-supervised GNNs focus on global characteristics, thus they employ encoders with global parameters and global objective functions. On the contrary, the proposed LRD-GNN pays much attention to local properties, and thus local encoder with a local objective function (low-rank decomposition) is utilized. The local encoder and objective function are essentially different from the global ones, and thus it is hard to point out their connections. Extensive experiments demonstrate the effectiveness and superiority of the local encoder and objective function. We want to conduct in-depth research on their theoretical connections in the future.
>
> ---
>
> ***Q2. It is difficult for readers, who are not familiar with low-rank, to get the key points in motivations and details of the proposed methods. It is better to review the concepts of low-rank and self-supervised in preliminaries section or appendix.***
>
> R2. Thanks for your suggestion. We will provide the basic concepts of low-rank and self-supervised GNNs in the appendix to improve the readability and completeness of the paper.
>
> ---
>
> ***Q3. There are some typos. For example, the is a nuclear norm missing in the formula between lines 162 and 163.***
>
> R3. Thanks for your suggestion. We will fully polish this paper. The formula between lines 162 and 163 can be corrected as
> $ \frac{1}{n_3} \sum_{i=1}^{n_3}  ||\bar{A}^{(i)}||_*$.
>
> ---
>
> ***Q4. The performance on preventing over-smoothing issue is compared against semi-supervised GNNs, i.e., GCN and GAT. The self-supervised GNNs should be compared with.***
>
> R4. According to your suggestion, the ability of our proposed LRD-GNN on preventing the over-smoothing issue is compared with DGI and MVGRL, which are representative self-supervised GNNs. The node classification results with various model depths on Cora, Citeseer, and Wiki-CS are as follows, as well as the visualization in Fig 1 in the global response PDF.  Therefore, compared to both semi-supervised and self-supervised GNNs, the proposed LRD-GNN can prevent over-smoothing issue.
>
> ***Table 1. The results on Cora.***
>  | Model   | 2-layers | 4-layers | 6-layers | 8-layers | 10-layers |
> |---------|----------|----------|----------|----------|-----------|
>  | DGI     | 82.15    | 74.47    | 54.25    | 31.22    | 27.63     |
>  | MVGRL   | 83.11    | 80.09    | 61.26    | 39.89    | 36.62     |
> | LRD-GNN | 84.74    | 81.77    | 81.62    | 80.53    | 80.42     |
>
> ***Table 2. The results on Citeseer.***
>  | Model   | 2-layers | 4-layers | 6-layers | 8-layers | 10-layers |
> |---------|----------|----------|----------|----------|-----------|
>  | DGI     | 70.72    | 64.02    | 44.78    | 25.47    | 24.16     |
> | MVGRL   | 71.51    | 65.43    | 62.24    | 57.7     | 47.59     |
>  | LRD-GNN | 71.94    | 71.82    | 70.8     | 69.32    | 68.36     |
>
> ***Table 2. The results on Wiki-CS.***
> | Model   | 2-layers | 4-layers | 6-layers | 8-layers | 10-layers |
> |---------|----------|----------|----------|----------|-----------|
>  | DGI     | 73.62    | 68.07    | 64.37    | 55.36    | 51.64     |
>  | MVGRL   | 76.53    | 72.66    | 68.27    | 60.36    | 52.54     |
>  | LRD-GNN | 81.55    | 81.17    | 80.62    | 79.53    | 79.42     |
>
> ---
>
> ***Q5.  The proposed LRD-GNN is a non-parameter model, which may benefit the task of self-supervised learning. However, how can it be improved if the labels are available?***
>
> R5. The proposed model can be improved by constructing the matrix and tensor with the help of node labels. The current self-supervised model constructs the matrix and tensor, which will be decomposed, using the ego-network and similar ego-networks without any supervision. When the node labels are available, the matrix and tensor constructions can be improved according to the predicted node labels. Specifically, the labels of nodes, which are unlabelled, can be predicted using the existing GNNs. Then, the matrix can be constructed using the nodes in the ego-network, which share the same predicted labels as center nodes. And the tensor can be constructed using the ego-networks, whose center nodes’ labels are the same. By employing the predicted node label, the nodes in the constructed matrix and tensor may belong to the same classes with higher probability, thus their ranks tend to be much lower. This facilitates the following decomposition. Therefore, the model can be improved with the help of node labels.

---

> ### Comment · Reviewer_EJvt · 2023-08-17
> **Response to the rebuttal**
>
> Thanks to the efforts made by the authors, all my concerns have been addressed. Therefore, I will maintain my acceptance of this paper.

---

### Official Review · Reviewer_7YeZ · 2023-07-06

**Soundness:** 3 good
**Presentation:** 2 fair
**Contribution:** 3 good
**Rating:** 7
**Confidence:** 4

**Summary:**

This paper proposes to alleviate the issues in propagation-based self-supervised GNN using the low-rank matrix/tensor decompositions. Firstly, it points out existing self-supervised GNNs can not capture local information and handle networks with heterophily due to the global learnable parameters and the lack of supervision information.  Secondly, it investigates the low-rank property of the local representation matrix. Thirdly, to meet these requirements, it presents low-rank matrix/tensor decomposition, which possesses attractive characteristics. Experiments on real networks verify the superior performance and robustness to noises.

**Strengths:**

- The motivation makes sense. It is interesting and meaningful to reveal the issues in existing self-supervised GNNs. Due the shared global parameter and lack of label information, propagation-based encoders tend to have some drawbacks. It seems the first attempt to seek specific encoder for self-supervised GNNs.

- The technology is solid and novel. The observation that local representation matrix should be low-rank is insightful and interesting. The employed low-rank matrix/tensor decomposition are novel to the field of graph neural network. Especially, the tensor decomposition seems technically solid. Besides, the proposed methods have some attractive properties.

- The evaluations are sufficient. Both the employed datasets and the baselines are representative and adequate. The performance improvements are acceptable. The visualization and ablation study are convincing.

**Weaknesses:**

- Some descriptions are not clear. For example, it is confusing to put the overflows of both matrix decomposition and tensor one in the same figure. Besides, Figure is too small and compact. See questions bellow.

-	It is difficult to understand why the proposed methods is better than propagation-based ones for reviewers who are not familiar with low-rank decomposition. It is not as intuitive as in computer vision. Therefore, some preliminaries should be given.

-	A section on related work should be given to make the readers to judge the contributions of this paper.

-	The presentation and writing should be carefully check.

**Questions:**

- What are the differences between unsupervised learning and self-supervised learning.  Although I like the idea of seeking representation using low-rank decomposition, I think it is an unsupervised learning. It may be different from self-supervised one. Could you explain their differences.

- The procedure of tensor construction is not clear. How the similar ego-networks are selected. Besides, it is not clear why the similar ego-networks are concatenated into a tensor instead of a larger matrix.

- It is not clear why the proposed methods are robust to topology and attribute noises. Although the experiments verify this characteristic, the formal explanation should be given.

- What do the white boxes mean in Fig 1(d)?

===

I have read the rebuttal and would like to keep my rating.

---

> ### Author Rebuttal · Authors · 2023-08-09
>
> ***Q1. What are the differences between unsupervised learning and self-supervised learning. Although I like the idea of seeking representation using low-rank decomposition, I think it is an unsupervised learning. It may be different from self-supervised one. Could you explain their differences.***
>
> R1. In my opinion, self-supervised learning belongs to unsupervised learning. Unsupervised learning is a broad class of tasks where the supervision information, such as node labels in the graph, is unavailable. It includes many tasks, such as clustering, dimension reduction, Probabilistic Density Estimation, etc. Self-supervised learning often employs to denote the unsupervised representation task with neural networks. Thus, self-supervised learning is a specific class of unsupervised learning. From this perspective, although low-rank decomposition is an unsupervised learning method, the proposed LRD-GNN is a self-supervised method for graph representation learning.
>
> ---
>
> ***Q2. The procedure of tensor construction is not clear. How the similar ego-networks are selected. Besides, it is not clear why the similar ego-networks are concatenated into a tensor instead of a larger matrix.***
>
> R2. Due to the limited space, the details on similar ego-networks selection are elaborated in the supplement material as follows. We construct the attribute tensor by selecting similar ego-networks and splicing the attribute matrices of similar ego-networks into a 3-way tensor. We evaluate similarity from two perspectives. (1) We select nodes that have similar attributes to the target node.  For instance, Cosine Similarity is used to measure the similarity of attributes between nodes. Then, the ego-networks from these nodes are selected to construct a tensor. (2) We select nodes that have similar local structures to the target node. The Shannon entropy value of ego-network is used to measure local structure similarity. For the ego-network $G_i = (V_i,E_i) $   around $v_i$ , the Shannon entropy value $H(G_i)$ is defined as $ H( G_i ) = -\sum_{v \in {V}_i} P(v)\log P(v) $, where $P(v)$ is the probability of the random walking visiting $v$ in ego-network.  Then, we choose several nodes' ego-networks that are close to the Shannon entropy value of the central node to construct a tensor.
>
> The reason for employing a tensor instead of a large matrix is because that tensor can preserve more structure information compared to the matrix. According to the construction of the tensor, nodes belonging to the same ego-network are on the same slice of the tensor, while nodes in different ego-networks are on different slices. On the contrary, the large matrix can not distinguish whether nodes are from the same ego-network, since all nodes are concatenated in the same dimension.  Therefore, the tensor is superior to the matrix.
>
> ***Q3. It is not clear why the proposed methods are robust to topology and attribute noises. Although the experiments verify this characteristic, the formal explanation should be given.***
>
> R3. The robustness to topology and attribute noises can be ascribed to the employed low-rank decomposition. On one hand, low-rank decomposition avoids the propagation over the topology, which is sensitive to the topology noises. Existing propagation-based GNNs perform the message passing according to the links in the topology. Therefore, the noisy links inevitably introduce noises in representation. On the contrary, the representation learning in LRD-GNN does not strictly rely the topology. The topology is only utilized to construct the local information matrix in the LRD-GNN, while the low-rank decomposition does not use topology. Therefore, even though the noisy topology introduces noises in the local information matrix, the following low-rank decomposition possesses the ability to denoise.  On another hand, the attribute noises can be removed via low-rank decomposition. The low-rank decomposition, which decomposes the attribute matrix into a low-rank matrix and a noisy one, has significant robustness to noises. Therefore, the low-rank decomposition contributes to the robustness to topology and attribute noises.
>
> ---
>
> ***Q4. What do the white boxes mean in Fig 1(d)?***
>
> R4. The white boxes stand for elements with zero values. In the Noise part, the discrete white boxes denote the elements without noises compared to the gray boxes, which represent the noises. In other parts, including Feature Matrices Extraction, Node Attribute, and Low-rank Matrix Representation, the row-wise white boxes stand for the padding vectors in the slices with fewer nodes. Specifically, the selected similar ego-networks contain different numbers of nodes, thus the constructed matrices possess different sizes. To concatenate these matrices into a tensor, matrices with fewer rows, i.e., ego-networks with fewer nodes, should be padded with zero rows. We will add this explanation to the caption of the figure in the final version.
>
> ---
>
> ***Weakness: 1) Figure is too small and compact. 2) Some preliminaries on low-rank should be given. 3) A section on related work should be given. 4) The presentation and writing should be carefully check.***
>
> R5.  Thanks for your suggestion on improving the paper. We will add the preliminaries on the low-rank problem and a related work section, fully polish the paper, and separate Figure 1 into the LRD-GNN-Martix Figure and the LRD-GNN-Tensor one in the appendix for better illustration.

---

### Official Review · Reviewer_Efzd · 2023-07-25

**Soundness:** 3 good
**Presentation:** 3 good
**Contribution:** 3 good
**Rating:** 6
**Confidence:** 3

**Summary:**

This paper studies the self-supervised graph learning for node classification tasks.
The authors make an observation that traditional propagation based GNNs loose discriminative information
via node property averaging. To address this issue, the authors propose a novel LRD-GNN method that
encourages low rank property of the representation matrix. The extended version, LRD-GNN-Tensor, allows
node feature propagation between long distance nodes and improve the method's ability to capture long
distance relationships. Empirical results show the proposed method outperforms baseline methods and improves
the model robustness towards noisy edges.

**Strengths:**

* The authors make an interesting observation that in order to propagate only among same class nodes, the
property matrix needs to be low rank. However, instead of enforcing a low rank decomposition of the
node property matrix, the method adopts the Robust PCA relaxation of the rank function/L0 norm and therefore
does not need heuristic hyper-parameters or label supervision.

* Empirical results show the proposed method has better performance than various baseline methods in both homophilic
and heterophilic settings. Ablation study also shows the proposed method can alleviate the over-smoothing issue.



**Weaknesses:**

* LRD-GNN-Tensor considers long distance correlations in the graph by grouping similar ego-networks together. However, it is not clear from the paper how are these similar ego-networks selected.

* It is claimed in the paper that the proposed method is scalable yet only small datasets are used in empirical studies.


**Questions:**

* Justification for selecting exactly M similar ego-networks for every node?

* How are the similar ego-networks selected in practice?

* Computation complexity and overall method scalability w.r.t. graph size?

**Limitations:**

The limitation of this work is not discussed in the paper.

I don't see any potential negative societal impact of this work.

---

> ### Author Rebuttal · Authors · 2023-08-09
>
> ***Q1. Justification for selecting exactly M similar ego-networks for every node?***
>
> R1. Thanks for your insightful question. Selecting exactly M similar ego-network is the compromise of the model’s expressive power and generalization ability. It is natural that different nodes should select different numbers of similar ego-networks, since different nodes possess different topology structures. Unfortunately, it requires additional components and parameters to determine it for every node. Although different M for different nodes can improve the model’s expressive power, it may induce an overfitting issue and reduce the generalization ability, especially in self-supervised learning tasks. Besides, we don’t find a practical relationship between the number of similar ego-networks and the node’s topology structure. This provides difficulty in modeling the number of similar ego-networks. From these perspectives, the strategy of exactly M similar ego-networks for every node is employed in our model.
>
> ---
>
> ***Q2. How are the similar ego-networks selected in practice?***
>
> R2. In practice, the cosine similarity of the center nodes’ attributes is utilized to select the similar ego-network. In the supplemental material, two similar ego-networks selection strategies, i.e., attribute-based strategy and topology-based one, are provided. The attribute-based strategy employs the cosine similarity of center nodes’ attributes as measurement, while the topology-based strategy uses the Shannon entropy value as measurement. The ablation study on the strategies is given in Section 4.1.3 and Table 4. It shows that the attribute-based strategy consistently outperforms the topology-based one. Therefore,  the cosine similarity of the center nodes’ attributes can be employed in practice.
>
> ---
>
> ***Q3. Computation complexity and overall method scalability w.r.t. graph size?***
>
> R3. The overall complexity is linear with the graph size, thus the model is scalable. The main component of the proposed LDR-GNN is the low-rank matrix/tensor decompositions, which are implemented via RPCA and tensor RPCA. The complexity of RPCA on a matrix of $n_1 \times n_2$, where $n_1 > n_2$, is $O(n_1 n_2^2)$, while that of tensor RPCA on a tensor of of $n_1 \times n_2 \times n_3$, where $n_1 > n_2$ and $n_3$ is the third dimension is $O(n_1 n_2^2 n_3 + n_1 n_2 n_3 log(n_3) )$. In the case of LDR-GNN, $n_3 = M$ is the number of selected similar ego-networks. Since the dimension of node attribute $F$ is often larger than the size of ego-network $d_i$, we set $n_1 = F$ and $n_2 = \bar{d}$, where $\bar{d}$ is the average node degree. Therefore, the complexities of LRD-GNN-Matrix and LRD-GNN-Tensor for each node are $O(F \bar{d}^2)$ and $O(F  \bar{d}^2  M + F  \bar{d}  M log(M) )$, respectively.  Since every node separately performs LRD-GNN-Matrix or LRD-GNN-Tensor, the overall complexities are $O(N F \bar{d}^2)$ and $O(N F  \bar{d}^2  M + N F  \bar{d}  M log(M) )$, where $N$ is the number of node in the graph, respectively. Therefore, the overall complexity is linear with the graph size, and the model is scalable.

---

> > ### Comment · Reviewer_Efzd · 2023-08-21
> >
> > I would like to thank the authors for their detailed rebuttal. All of my questions are answered. However, as the weaknesses I listed above still hold, I will keep my score.

---

> > > ### Author Response · Authors · 2023-08-22
> > > **Response to the Weakness**
> > >
> > > Thanks for your feedback.  For the two weaknesses, we would like to clarify as follows.
> > >
> > >  **W1. Similar ego-networks selection.**
> > >
> > > R1.  The procedure of similar ego-network selection has been in the submitted appendix, while the ablation study has been performed in the experiment section.  Besides, this procedure has also been clarified in the response to Q2.
> > >
> > > ---
> > >
> > > **W2.  Only small datasets are employed.**
> > >
> > > R2.  To the best of our knowledge, almost all datasets for node-level self-supervised learning have been used in this paper. Thus, according to the complexity analysis in the response to Q3, the proposed method is scalable to large graphs.
> > >
> > >
> > > **Therefore, we hope the weaknesses can be clarified.**

---

### Author Rebuttal · Authors · 2023-08-09

We would like to express our sincere appreciations to the reviewers for their insightful comments and compliments to our paper. The PDF contains the figure of results on preventing over-smoothing issue.

---

### Comment · Area_Chair_Hmqs · 2023-08-18
**Please read the rebuttals and add a comment to acknowledge**

Dear Reviewers,

Many thanks for your time and efforts on this paper. As the end of discussion is coming soon, please read the rebuttal to see if your concerns/questions are properly resolved and add a comment to acknowledge that you have read the rebuttal.

Many Thanks.

---

### Decision · Program_Chairs · 2023-09-21

**Decision:**

Accept (poster)

**Comment:**

All the reviewers agree that this is a well-written paper with novel approaches to encourage low-rank property for GNN in the semi-supervised setting. Empirical results also demonstrated superior performance compared to other methods. I recommend to accept this paper.